# Learning safe policies with expert guidance

**Jessie Huang**[1]    **Fa Wu**[1][2]    **Doina Precup**[1]    **Yang Cai**[1]
[1]School of Computer Science, McGill University
[2]Zhejiang Demetics Medical Technology
{jiexi.huang,fa.wu2}@mcgill.ca, {dprecup,cai}@cs.mcgill.ca

## Abstract

We propose a framework for ensuring safe behavior of a reinforcement learning agent when the reward function may be difficult to specify. In order to do this, we rely on the existence of demonstrations from expert policies, and we provide a theoretical framework for the agent to optimize in the space of rewards consistent with its existing knowledge. We propose two methods to solve the resulting optimization: an exact ellipsoid-based method and a method in the spirit of the "follow-the-perturbed-leader" algorithm. Our experiments demonstrate the behavior of our algorithm in both discrete and continuous problems. The trained agent safely avoids states with potential negative effects while imitating the behavior of the expert in the other states.

## 1   Introduction

In Reinforcement Learning (RL), agent behavior is driven by an objective function defined through the specification of rewards. Misspecified rewards may lead to *negative side effects* [3], when the agent acts unpredictably responding to the aspects of the environment that the designer overlooked, and potentially causes harms to the environment or itself. As the environment gets richer and more complex, it becomes more challenging to specify and balance rewards for every one of its aspects. Yet if we want to have some type of safety guarantees in terms of the behavior of an agent learned by RL once it is deployed in the real world, it is crucial to have a learning algorithm that is robust to mis-specifications.

We assume that the agent has some knowledge about the reward function either through past experience or demonstrations from experts. The goal is to choose a robust/safe policy that achieves high reward with respect to any reward function that is consistent with the agent's knowledge [1]. We formulate this as a *maxmin learning* problem where the agent chooses a policy and an adversary chooses a reward function that is consistent with the agent's current knowledge and minimizes the agent's reward. The goal of the agent is to learn a policy that maximizes the worst possible reward.

We assume that the reward functions are linear in some feature space. Our formulation has two appealing properties: (1) it allows us to combine demonstrations from multiple experts even though they may disagree with each other; and (2) the training environment/MDP in which the experts operate need not be the same as the testing environment/MDP where the agent will be deployed, our results hold as long as the testing and training MDPs share the same feature space. As an application, our algorithm can learn a maxmin robust policy in a new environment that contains a few features that are not present in the training environment. See our gridworld experiment in Section 5.

Our first result (Theorem 1) shows that given any algorithm that can find the optimal policy for an MDP in polynomial time, we can solve the maxmin learning problem exactly in polynomial

time. Our algorithm is based on a seminal result from combinatorial optimization – the equivalence between separation and optimization [9, 14] – and the ellipsoid method. To understand the difficulty of our problem, it is useful to think of maxmin learning as a two-player zero-sum game between the agent and the adversary. The deterministic policies correspond to the pure strategies of the agent. The consistent reward functions we define in Section 3 form a convex set and the adversary's pure strategies are the extreme points of this convex set. Unfortunately, both the agent and the adversary may have exponentially many pure strategies, which are hard to describe explicitly. This makes solving the two-player zero-sum game challenging. Using tools from combinatorial optimization, we manage to construct separation oracles for both the agent's and the adversary's set of policies using the MDP solver as a subroutine. With the separation oracles, we can solve the maxmin learning problem in polynomial time using the ellipsoid method.

Theorem 1 provides a polynomial time algorithm, but as it heavily relies on the ellipsoid method, it is computationally expensive to run in practice. We propose another algorithm (Algorithm 3) based on the online learning algorithm – *followed-the-perturbed-leader (FPL)*, and show that after $T$ iterations the algorithm computes a policy that is at most $O\left(1/\sqrt{T}\right)$ away from the true maxmin policy (Theorem 2). Moreover, each iteration of our algorithm is polynomial time. Notice that many other low-regret learning algorithms, such as the multiplicative weights update method (MWU), are not suitable for our problem. The MWU requires explicitly maintaining a weight for every pure strategy and updates them in every iteration, resulting in an exponential time algorithm for our problem. Furthermore, we show that Algorithm 3 still has similar performance when we only have a fully polynomial time approximation scheme (FPTAS) for solving the MDP. The formal statement and proof are postponed to the supplemental material due to space limit.

## 1.1 Related Work

In the sense of using expert demonstrations, our work is related to inverse reinforcement learning (IRL) and apprenticeship learning [18, 1, 21, 20]. In particular, the apprenticeship learning problem aims to learn a policy that performs at least as well as the expert's policy on all basis rewards, and can also be formulated as a maxmin problem [21, 20]. Despite the seemingly similarity, our maxmin learning problem aims to solve a completely different problem than apprenticeship learning. Here is a simple example: consider in a gridworld, there are two basis rewards, $w_1$ and $w_2$, and there are only two routes/policies – *top* and *bottom*. The expert takes the *top* route, getting 100 under $w_1$ and 70 under $w_2$. Alternatively, taking the *bottom* route gets 90 under both $w_1$ and $w_2$. Apprenticeship learning will return the *top* route, because taking the alternative route performs worse than the expert under $w_1$ and violates the requirement. What is our solution? Assuming $\epsilon = 25$ [2], both $w_1$ and $w_2$ (or any convex combination of them) are consistent with the expert demonstration. If we choose the top route, the worst case performance (under $w_2$ in this case) is 70, while the correct maxmin solution to our problem is to take the *bottom* route so that its worst performance is 90. In the worst case (under $w_2$), our maxmin policy has better guarantees and thus is more robust. Unlike apprenticeship learning/IRL, we do not want to mimic the experts or infer their rewards, but we want to produce a policy with robustness guarantees by leveraging their data. As a consequence, our results are applicable to settings where the training and testing environments are different (as discussed in the Introduction). Moreover, our formulation allows us to combine multiple expert demonstrations.

Inverse reward design [10] uses a proxy reward and infers the true reward by estimating its posterior. Then it uses risk-averse planning together with samples from the posterior in the testing environment to achieve safe exploration. Our approach achieves a similar goal without assuming any distribution over the rewards and is arguably more robust. We apply a single reward function to the whole MDP while they apply (maybe too pessimistically) per step/trajectory maxmin planning. Furthermore, our algorithm is guaranteed to find the maxmin solution in polynomial time, and can naturally accommodate multiple experts.

In repeated IRL [2], the agent acts on the behalf of a human expert in a variety of tasks, and the human expert corrects the agent when the agent's policy is far from the optimum. The goal is to minimize the number of corrections from the expert, and they provide an upper bound on the number of corrections by reducing the problem to the ellipsoid method. Their model requires continuous interaction with an expert while our model only assumes the availability of one or a couple expert

policies prior to training. Furthermore, we aim to find a maxmin optimal policy, while their paper focuses on minimizing the number of corrections needed.

Robust Markov Decision Processes [19, 12] have addressed the problem of performing dynamic programming-style optimization environments in which the transition probability matrix is uncertain. Lim, Xu & Mannor [16] have extended this idea to reinforcement learning methods. This body of work also uses min-max optimization, but because the optimization is with respect to worst-case transitions, this line of work results in very pessimistic policies. Our algorithmic approach and flavor of results are also different. [17] have addressed a similar adversarial setup, but in which the environment designs a worst-case disturbance to the dynamics of the agent, and have addressed this setup using $H_\infty$ control.

**Paper Organization:** We introduce the notations and define the maxmin learning problem in Section 2. We provide three different ways to define the set of consistent reward functions in Section 3, and present the ellipsoid-based exact algorithm and its analysis in Section 4.1. The FPL-based algorithm and its analysis are in Section 4.2, followed by experimental results in Section 5.

## 2 Preliminary

An MDP is a tuple $M = (\mathcal{S}, \mathcal{A}, P_{sa}, \gamma, D, R)$, including a finite set of states, $\mathcal{S}$, a set of actions, $\mathcal{A}$, and transition probabilities, $P_{sa}$. $\gamma$ is a discount factor, and $D$ is the distribution of initial states. The reward function $R$ instructs the learning process. We assume that the reward is a linear function of some vector of features $\phi: \mathcal{S} \to [0, 1]^k$ over states. That is $R(s) = w \cdot \phi(s)$ for every state $s \in \mathcal{S}$, where $w \in \mathbb{R}^k$ is the *reward weights* of the MDP. The true reward weights $w^*$ is assumed to be unknown to the agent. We use $\langle \cdot \rangle$ to denote the bit complexity of an object. In particular, we use $\langle M \rangle$ to denote the bit complexity of $M$, which is the number of bits required to represent the distribution of initial states, transition probabilities, the discount factor $\gamma$, and the rewards at all the states. We use the notation $M \backslash R$ to denote a MDP without the reward function, and $\langle M \backslash R \rangle$ is its bit complexity. We further assume that $\phi(s)$ can be represented using at most $\langle \phi \rangle$ bits for any state $s \in \mathcal{S}$.

An agent selects the action according to a policy $\pi$. The value of a policy under rewards $w$ is $\mathbb{E}_{s_0 \sim D}[V^\pi(s_0)|M] = w \cdot \mathbb{E}[\sum_{t=0}^\infty \gamma^t \phi(s_t)|M, \pi]$. It is expressed as the weights multiplied by the accumulated discounted feature value given a policy, which we define as $\Psi(\pi) = \mathbb{E}[\sum_{t=0}^\infty \gamma^t \phi(s_t)|M, \pi]$.

**MDP solver** We assume that there is a RL algorithm ALG that takes an MDP as input and outputs an optimal policy and its corresponding representation in the feature space. In particular, $\text{ALG}(M)$ outputs $(\pi^*, \mu^*)$ such that $\mathbb{E}_{s_0 \sim D}[V^{\pi^*}(s_0)|M] = \max_\pi \mathbb{E}_{s_0 \sim D}[V^\pi(s_0)|M]$ and $\mu^* = \Psi(\pi^*)$.

**Maxmin Learning** All weights that are consistent with the agent's knowledge form a set $P_R$. We will discuss several formal ways to define this set in Section 3. The goal of the agent is to learn a policy that maximizes the reward for any reward function that could be induced by weights in $P_R$ and adversarially chosen. More specifically, the max-min learning problem is $\max_{\mu \in P_F} \min_{w \in P_R} w^T \mu$, where $P_F$ is the polytope that contains the representations of all policies in the feature space, i.e. $P_F = \{\mu \mid \mu = \Psi(\pi) \text{ for some policy } \pi \}$. WLOG, we assume that all weights lie in $[-1, 1]^k$.

**Separation Oracles** To perform maxmin learning, we often need to optimize linear functions over convex sets that are intersections of exponentially many halfspaces. Such optimization problem is usually intractable, but if the convex set permits a polynomial time *separation oracle*, then there exists polynomial time algorithms (e.g. ellipsoid method) that optimize linear functions over it.

**Definition 1.** *(Separation Oracle) Let $P$ be a closed, convex subset of Euclidean space $\mathbb{R}^d$. Then a* Separation Oracle *for $P$ is an algorithm that takes as input a point $x \in \mathbb{R}^d$ and outputs "YES" if $x \in P$, or a hyperplane $(w, c)$ such that $w \cdot y \leq c$ for all $y \in P$, but $w \cdot x > c$. Note that because $P$ is closed and convex, such a hyperplane always exists whenever $x \notin P$.*

## 3 Consistent Reward Polytope

In this section, we discuss several ways to define the consistent reward polytope $P_R$.

**Explicit Description** We assume that the agent knows that the weights satisfy a set of explicitly defined linear inequalities of the form $c \cdot w \geq b$. For example, such an inequality can be learned by observing that a particular policy yields a reward that is larger or smaller than a certain threshold. [3]

**Implicitly Specified by an Expert Policy** Usually, it may not be easy to obtain many explicit inequalities about the weights. Instead, we may have observed a policy $\pi_E$ used by an expert. We further assume that the expert's policy has a reasonably good performance under the true rewards $w^*$. Namely, $\pi_E$'s expected reward is only $\epsilon$ less than the optimal one. Let the expert's feature vector $\mu_E = \Psi(\pi_E)$. The set $P_R$ therefore contains all $w$ such that $\mu_E \cdot w \geq \mu^T \cdot w - \epsilon, \forall \mu \in P_F$. It is not hard to verify that under this definition $P_R$ is a convex set. Even though explicitly specifying $P_R$ is extremely expensive as there are infinitely many $\mu \in P_F$, we can construct a polynomial time separation oracle $SO_R$ (Algorithm 1). An alternative way to define $P_R$ is to assume that the expert policy can achieve $(1 - \epsilon)$ of the optimal reward (assuming the final reward is positive). We can again design a polynomial time separation oracle similar to Algorithm 1.

---

**Algorithm 1** Separation Oracle $SO_R$ for the reward polytope $P_R$

---

**input** $w' \in \mathbb{R}^k$
1: Let $\mu_{w'} := \text{argmax}_{\mu \in P_F} \mu \cdot w'$. Notice that $\mu_{w'}$ is the feature vector of the optimal policy under reward weights $w'$. Hence, it can be computed by our MDP solver ALG.
2: **if** $\mu_{w'} \cdot w' > \mu_E \cdot w' + \epsilon$ **then**
3:     output "NO" , and $(\mu_E - \mu_{w'}) \cdot w + \epsilon \geq 0$ as the separating hyperplane, since for all $w \in P_R, \mu_E \cdot w \geq \mu_{w'} \cdot w - \epsilon$.
4: **else**
5:     output "YES".
6: **end if**

---

**Combining Multiple Experts** How can we combine demonstrations from experts operating in drastically different environments? Here is our model. For each environment $i$, there is a separate MDP $M_i$, and all the MDPs share the same underlying weights as they are all about completing the same task although in different environments. The $i$-th expert's policy is nearly optimal in $M_i$. More specifically, for expert $i$, her policy $\pi_{E_i}$ is at most $\epsilon_i$ less than the optimal policy in $M_i$. Therefore, each expert $i$ provides a set of constraints that any consistent reward needs to satisfy, and $P_R$ is the set of rewards that satisfy all constraints imposed by the experts. For each expert $i$, we can design a separation oracle $SO_R^{(i)}$ (similar to Algorithm 1) accepting weights that respect the constraints given by expert $i$'s policy. We can easily design a separation oracle for $P_R$ that only accepts weights that will be accepted by all separation oracles $SO_R^{(i)}$.

From now on, we will not distinguish between different ways to define and access the consistent reward polytope $P_R$, but simply assume that we have a polynomial time separation oracle for it. All the algorithms we design in this paper only require access to this separation oracle. In Section 5, we will specify how the $P_R$ is defined for each experiment.

## 4 Maxmin Learning using an Exact MDP Solver

In this section, we show how to design maxmin learning algorithms. Our algorithm only interacts with the MDP through the MDP solver, which can be either model-based or model-free. Our first algorithm solves the maxmin learning problem exactly using the ellipsoid method. Despite the fact that the ellipsoid method has provable worst-case polynomial running time, it is known to be inefficient sometimes in practice. Our second algorithm is an efficient iterative method based on the online learning algorithm – *follow-the-perturbed-leader (FPL)*.

### 4.1 Ellipsoid-Method-Based Solution

**Theorem 1.** *Given a polynomial time separation oracle $SO_R$ for the consistent reward polytope $P_R$ and an exact polynomial time MDP solver* ALG*, we have a polynomial time algorithm such that*

*for any MDP without the reward function $M\backslash R$, the algorithm computes the maxmin policy $\pi^*$ with respect to $M\backslash R$ and $P_R$.*

The plan is to first solve the maxmin learning problem in the feature space then convert it back to the policy space. Solving the maxmin learning problem in the feature space is equivalent to solving the linear program in Figure 1.

$$
\begin{aligned}
\max \quad & z \\
\text{subject to} \quad & z \le \mu \cdot w, \quad \forall w \in P_R \\
& \mu \in P_F
\end{aligned}
$$

Figure 1: Maxmin Learning LP.

The challenges for solving the LP are that (i) it is not clear how to check whether $\mu$ lies in the polytope $P_F$, and (ii) there are seemingly infinitely many constraints of the type $z \le \mu \cdot w$ as there are infinitely many $w \in P_R$. Next, we show that given an exact MDP solver ALG, we can design a polynomial time separation oracle for the set of feasible variables $(\mu, z)$ of LP 1. With this separation oracle, we can apply the ellipsoid method (see Theorem 3 in the supplementary material) to solve LP 1 in polynomial time.

First, we design a separation oracle for polytope $P_F$ by invoking a seminal result from optimization – the equivalence between separation and optimization.

**Lemma 1** (Separation ≡ Optimization). *[9, 14] Consider any convex polytope $P = \{x : Ax \le b\} \in \mathbb{R}^d$ and the following two problems:*

&emsp;***Linear Optimization:*** *given a linear objective $c \in \mathbb{R}^d$, compute $x^* \in argmax_{x \in P} c \cdot x$*

&emsp;***Separation:*** *given a point $y \in \mathbb{R}^d$, decide that $y \in P$, or else find $h \in \mathbb{R}^d$ s.t. $h \cdot x < h \cdot y, \forall x \in P$.*

*If $P$ can be described implicitly using $\langle P \rangle$ bits, then the separation problem is solvable in $poly(\langle P \rangle, d, \langle y \rangle)$ time for $P$ if and only if the linear optimization problem is solvable in $poly(\langle P \rangle, d, \langle c \rangle)$ time.*

It is not hard to see that if one can solve the separation problem, one can construct a separation oracle in polynomial time and apply the ellipsoid method to solve the linear optimization problem. The less obvious direction in the result above states that if one can solve the linear optimization problem, one can also use it to construct a separation oracle. The equivalence between these two problems turns out to have profound implications in combinatorial optimization and has enabled numerous polynomial time algorithms for many problems that are difficult to solve otherwise.

---

**Algorithm 2** Separation Oracle for the feasible $(\mu, z)$ in LP 1

**input** $(\mu', z') \in \mathbb{R}^{k+1}$
1: Query $SO_F(\mu')$.
2: **if** $\mu' \notin P_F$ **then**
3: &emsp; output "NO" and output the same separating hyperplane as outputted by $SO_F(\mu')$.
4: **else**
5: &emsp; Let $w^* \in argmin_{w \in P_R} \mu' \cdot w$ and $V = \mu' \cdot w^*$. This requires solving a linear optimization problem over $P_R$ using the ellipsoid method with the separation oracle $SO_R$.
6: &emsp; **if** $z' \le V$ **then**
7: &emsp;&emsp; output "YES"
8: &emsp; **else**
9: &emsp;&emsp; output "NO", and a separating hyperplane $z \le \mu \cdot w^*$, as $z' > \mu' \cdot w^*$ and all feasible solutions of LP 1 respect this constraint.
10: &emsp; **end if**
11: **end if**

---

Our goal is to design a polynomial time separation oracle for the polytope $P_F$. The key observation is that the linear optimization problem over polytope $P_F$: $\max_{\mu \in P_F} w \cdot \mu$ is exactly the same as solving the MDP with reward function $R(\cdot) = w \cdot \phi(\cdot)$. Therefore, we can use the MDP solver to design a polynomial time separation oracle for $P_F$.

**Lemma 2.** *Given access to an MDP solver* ALG *that solves any MDP $M$ in time polynomial in $\langle M \rangle$, we can design a separation oracle $SO_F$ for $P_F$ that runs in time polynomial in $\langle M\backslash R \rangle$, $\langle \phi \rangle$, $k$, and the bit complexity of the input.[4]*

With $SO_F$, we first design a polynomial time separation oracle for checking the feasible $(z, \mu)$ pairs in LP 1 (Algorithm 2). With the separation oracle, we can solve LP 1 using the ellipsoid method. The last difficulty is that the optimal solution only gives us the maxmin feature vector instead of the corresponding maxmin policy. We use the following nice property of $SO_F$ to convert the optimal solution in the feature space to the policy space. See Section 8 in the supplementary material for intuition behind Lemma 3.

**Lemma 3.** *[9, 14, 7] If $SO_F(\mu) = $ "YES", there exists a set, $C$, of weights $w \in \mathbb{R}^k$ such that $SO_F$ has queried the MDP solver ALG on reward function $w \cdot \phi(\cdot)$ for every $w \in C$. Let $(\pi_w, \mu_w)$ be the output of ALG on weight $w$, then $\mu$ lies in the convex hull of $\{\mu_w | w \in C\}$.*

*Proof of Theorem 1*: It is not hard to see that Algorithm 2 is a valid polynomial time separation oracle for the feasible $(\mu, z)$ pairs in LP 1. Hence, we can solve LP 1 in polynomial time with the ellipsoid method with access to Algorithm 2. Next, we show how to convert the optimal solution $\mu^*$ of LP 1 to the corresponding maxmin optimal policy $\pi^*$. Here, we invoke Lemma 3. We query $SO_F$ on $\mu^*$ and we record all weights $w$ that $SO_F$ has queried the MDP solver ALG on. Let $C = \{w_1, \ldots, w_\ell\}$ be all the queried weights. As $SO_F$ is a polynomial time algorithm, $\ell$ is also polynomial in the input size. By Lemma 3, we know that $\mu$ is in the convex hull of $(\{\mu_w | w \in C\})$, which means there exists a set of nonnegative numbers $p_1, \ldots, p_\ell$, such that $\sum_{i=1}^{\ell} p_i = 1$ and $\mu^* = \sum_{i=1}^{\ell} p_i \cdot \mu_{w_i}$. Clearly, the discounted accumulated feature value of the randomized policy $\sum_{i=1}^{\ell} p_i \cdot \pi_{w_i}$ equals to $\sum_{i=1}^{\ell} p_i \cdot \Psi(\pi_{w_i}) = \sum_{i=1}^{\ell} p_i \cdot \mu_{w_i} = \mu^*$. We can compute the $p_i$s in poly-time via linear programming and $\sum_{i=1}^{\ell} p_i \cdot \pi_{w_i}$ is the maxmin policy. $\square$

## 4.2 Finding the Maxmin Policy using Follow the Perturbed Leader

The exact algorithm of Theorem 1 may be computationally expensive to run, as the separation oracle $SO_F$ requires running the ellipsoid method to answer every query, and on top of that we need to run the ellipsoid method with queries to $SO_F$. In this section, we propose a simpler and faster algorithm that is based on the algorithm – *follow-the-perturbed-leader (FPL)* [13].

**Theorem 2.** *For any $\xi \in (0, 1/2)$, with probability at least $1 - 2\xi$, Algorithm 3 finds a policy $\pi$ after $T$ rounds of iterations such that its expected reward under any weight from $P_R$ is at least $\max_{\mu \in P_F} \min_{w \in P_R} \mu \cdot w - \frac{k^2 \left( 6 + 4\sqrt{\ln 1/\xi} \right)}{\sqrt{T}}$. In every iteration, Algorithm 3 makes one query to ALG and $O\left( k^2 \left( (\log k)^2 + ((b + \langle \phi \rangle)(|\mathcal{A}||\mathcal{S}| + k) + \log T)^2 \right) \right)$ queries to $SO_R$, where $b$ is an upper bound on the number of bits needed to specify the transition probability $P_{sa}$ for any state $s$ and action $a$.*

FPL is a classical online learning algorithm that solves a problem where a series of decisions $d_1, d_2, \ldots$ need to be made. Each $d_i$ is from a possibly infinite set $D \subseteq \mathbb{R}^n$. The state $s_t \in \mathbb{S} \subseteq \mathbb{R}^n$ at step $t$ is observed after the decision $d_t$. The goal is to have the total reward $\sum_t d_t \cdot s_t$ not far from the reward of the best fixed decision from $D$ in hindsight, that is $\max_{d \in D} \sum_t d \cdot s_t$. The FPL algorithm guarantees that after $T$ rounds, the regret $\sum_t d_t \cdot s_t - \max_{d \in D} \sum_t d \cdot s_t$ scales linearly in $\sqrt{T}$. This guarantee holds for both oblivious and adaptive adversary, and the bound holds both in expectation and with high probability (see Theorem 4 in Section 8 of the supplementary material for the formal statement).

FPL falls into a large class of algorithms that are called low-regret algorithms, as the regret grows sub-linearly in $T$. It is well known that low-regret algorithms can be used to solve two-player zero-sum games approximately. The maxmin problem we face here can also be modeled as a two-player zero-sum games. One player is the agent whose strategy is a policy $\pi$, and the other player is the reward designer whose strategy is a weight $w \in P_R$. The agent's payoff is the reward that it collects using policy $\pi$, which is $\Psi(\pi) \cdot w$, and the designer's payoff is $-\Psi(\pi) \cdot w$. Finding the maxmin strategy for the agent is equivalent to finding the maxmin policy. One challenge here is that the numbers of strategies for both players are infinite. Even if we only consider the pure strategies which correspond to the extreme points of $P_F$ and $P_R$, there are still exponentially many of them. Many low-regret algorithms such as multiplicative-weights-update requires explicitly maintaining a distribution over the pure strategies, and update it in every iteration. In our case, these algorithms will take exponential time to finish just a single iteration. This is the reason why we favor the FPL algorithm, as the FPL algorithm only requires finding the best policy giving the past weights, which

can be done by the MDP solver ALG. We also show that a similar result holds even if we replace the exact MDP solver with an additive FPTAS $\widehat{\text{ALG}}$. The proof of Theorem 2 can be found in Section 8 in the supplementary material. Our generalization to cases where we only have access to $\widehat{\text{ALG}}$ is postponed to Section 9 in the supplementary material.

---

**Algorithm 3** FPL Maxmin Learning

---
**input** $T$: the number of iterations
1: Set $\delta := 1/k\sqrt{T}$.
2: Arbitrarily pick some policy $\pi_1$, compute $\mu_1 \in P_F$. Arbitrarily pick some reward weights $w_1$, and set $t = 1$.
3: **while** $t \leq T$ **do**
4:    Use ALG to compute the optimal policy $\pi_t$ and $\mu_t = \Psi(\pi_t)$ that maximizes the expected reward under reward function $\left(\sum_{i=1}^{t-1} w_i + p_t\right) \cdot \phi(\cdot)$, where $p_t$ is drawn uniformly from $[0, 1/\delta]^k$.
5:    Let $w_t := \text{argmin}_{w \in P_R} w^T (\sum_{i=1}^{t-1} \mu_t + q_t)$, where $q_t$ is drawn uniformly from $[0, 1/\delta]^k$.
6:    $t := t + 1$.
7: **end while**
8: Output the randomized policy $\frac{1}{T} \cdot \sum_{t=1}^{T} \pi_t$.

---

## 5 Experiments

**Gridworld**   We use gridworlds in the first set of experiments. Each grid may have a different "terrain" type such that passing the grid will incur certain reward. For each grid, a feature vector $\phi(s)$ denotes the terrain type, and the true reward can be expressed as $R^* = w^* \cdot \phi(s)$. The agent's goal is to move to a goal grid with *maximal* reward under the *worst possible* weights that are consistent with the expert. In other words, the maxmin policy is a safe policy, as it avoids possible negative side effects [3]. In the experiments, we construct the expert policy $\pi_E$ in a small (10×10) demonstration gridworld that contains a subset of the terrain types. One expert policy is provide, and the number of trajectories that we need to estimate the expert policy's cumulative feature follows the sample complexity analysis as in [21]. In the following experiment we set $\epsilon = 0.5$ which defines $P_R$ and captures how close to optimal the expert is.

An example behavior is shown in Figure 3. There are 5 possible terrain types. The expert policy in Figure 3 (left) has only seen 4 terrain types. We compute the maxmin policy in the "real-world" MDP of a much larger size (50×50) with all 5 terrain types using Algorithm 3 with the reward polytope $P_R$ implicitly specified by the expert policy. Figure 3 (middle) shows that our maxmin policy avoids the red-colored terrain that was missing from the demonstration MDP. To facilitate observation, Figure 3 (right) shows the same behavior by an agent trained in a smaller MDP. Figure 2 compares the maxmin policy to a baseline. The baseline policy is computed in an MDP whose reward weights are the same as the demonstration MDP for the first four terrain types and the fifth terrain weight is chosen at random. Our maxmin policy is much safer than the baseline as it completely avoids the fifth terrain type. It also imitates the expert's behavior by favoring the same terrain types.

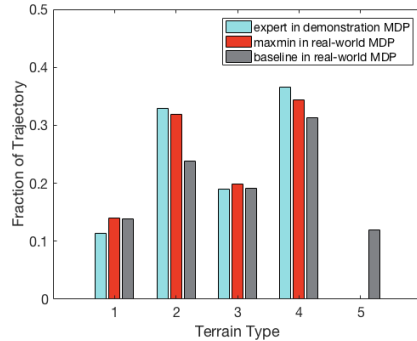

Figure 2: Experiment results comparing our maxmin policy to a baseline. The baseline was computed with a random reward for the fifth terrain and the other four terrain rewards set the same as the demonstration MDP. Our maxmin policy is much safer than the baseline as it completely avoids traversing the fifth (unknown) terrain type. It should also be noticed that the maxmin policy learns from the expert policy while achieving the goal of avoiding potential negative side effects, as the fraction of trajectory of each terrain type closely resemble the expert.

We also implemented the maxmin method in gridworlds with a stochastic transition model. The maxmin policy (see Figure 8 in Section 10

of the supplementary material) is more conservative comparing to the deterministic model, and chooses paths that are further away from any unknown terrains. More details and computation time can be found in the supplementary material.

It should be noted that the training and testing MDPs are different. More specifically, the red terrain type is missing from the expert demonstration, and the testing MDP is of a larger size. As discussed in the Introduction, our formulation allows the testing MDP in which the agent operates to be different from the training MDP in which the expert demonstrates, as long as the two MDPs share the same feature space. All of our experiments have this property. To the limit of our knowledge, apprenticeship learning requires the training and testing MDPs to be the same, thus a direct comparison is not possible. For example, in the gridworld experiments, one has to explicitly assign a reward to the "unknown" feature in order to apply apprenticeship learning, which may cause the problem of reward misspecification and negative side effects. Our maxmin solution is robust to such issues.

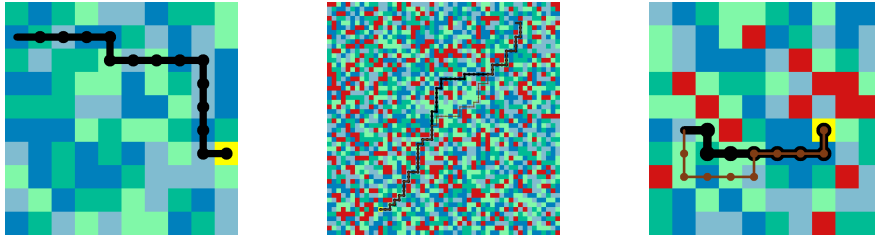

Figure 3: An example of maxmin policy in gridworlds. **Left**: an expert policy in the small demonstration MDP, where 4 of 5 terrain types were used and their weights were randomly chosen. The expert policy guides moving towards the yellow goal grid while preferring the terrains with higher rewards (light blue and light green). **Middle**: when faced with terrain types (red-colored) that the expert policy never experienced, maxmin policy avoids such terrains and the accompanying negative side effects. The agent learns to operate in a larger (50×50) grid world. **Right**: an agent in a smaller MDP to facilitate observation.The maxmin policy generates two possible trajectories.

**CartPole** Our next experiments are based on the classic control task of cartpole and the environment provided by OpenAI Gym [6]. While we can only solve the problem approximately using model-free learning methods, our experiments show that our FPL-based algorithm can learn a safe policy efficiently for a continuous task. Moreover, if provided with more expert policies, our maxmin learning method can easily accomodate and learn from multiple experts.

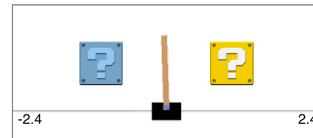

Figure 4: Modified cartpole task with two additional features – questions blocks to either side of the center. The rewards associated with passing these blocks are not provided to the agent.

We modify the cartpole problem by adding two possible features to the environment as the two question blocks shown in Figure 4, and more details in the supplementary material. The agent has no idea of what consequences passing these two blocks may have. Instead of knowing the rewards associated with these two blocks, we have expert policies from two other related scenarios. The first expert policy (*Expert A*) performs well in *scenario A* where only the blue block to the left of the center is present, and the second expert policy (*Expert B*) performs well in *scenario B* where only the yellow block to the right of the center is present. The behavior of expert policies in a default scenario (without any question blocks), and scenarios $A$ and $B$ are shown in Figure 5. It is obvious that comparing with the default scenario, the expert policies in the other two scenarios prefer to travel to the right side. Intuitively, it seems that the blue block incurs negative effects while the yellow block is either neutral or positive.

Now we train the agent in the presence of both question blocks. First, we provide the agent with *Expert A* policy alone, and learn a maxmin policy. The maxmin policy's behavior is shown in Figure 6 (**top**). It tries to avoid both question blocks since it observes that Expert A avoids the blue block and it has no knowledge of the yellow block. Then, we provide both $Expert\ A$ and $Expert\ B$ to the agent, and the resulting maxmin policy guides movement in a wider range extending to the right of

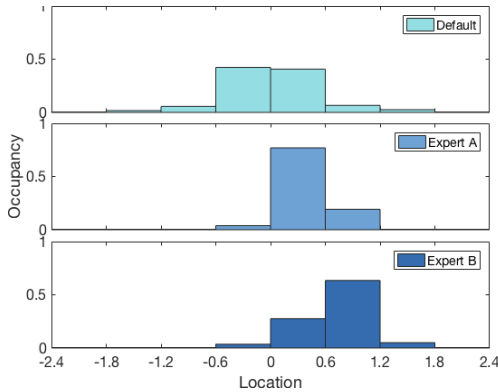

Figure 5: Behavior examples of different policies. Occupancy is defined as the number of steps appearing at a location divided by the total steps. **top**: In the default setting without any question blocks, the travel range is relatively symmetric around the center of the field. **mid**: In the presence of the blue question block to the left, an expert policy $A$ guides movements to the right. **bottom**: In *scenario B* where only the yellow question block is present, expert policy $B$ also guides movement to the right.

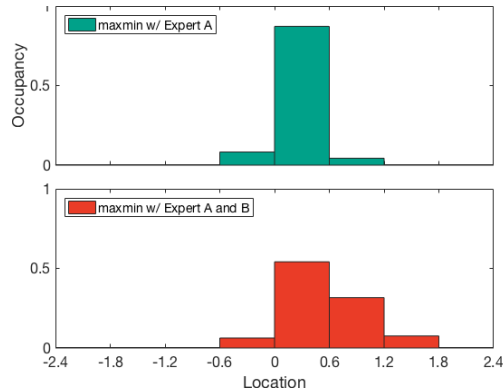

Figure 6: Maxmin policy learnt with different expert policies. **top**: Given *Expert A* policy only, the agent learns to stay within a narrow range near slightly right to the center to avoid both question blocks. Because the agent has no knowledge about the yellow block, a maxmin policy avoids it. **bottom**: When given both *Expert A* and *Expert B* policies, the agent learns that it is safe to pass the yellow block, so the range is wider and extends more to the right comparing to the maxmin policy learnt from *Expert A* alone.

the field as shown in Figure 6 (**bottom**). This time, our maxmin policy also learns from *Expert B* that the yellow block is not harmful. The experiment demonstrates that our maxmin method works well with complex reinforcement learning tasks where only approximate MDP solvers are available.

## 6 Discussion

In this paper, we provided a theoretical treatment of the problem of reinforcement learning in the presence of mis-specifications of the agent's reward function, by leveraging data provided by experts. The posed optimization can be solved exactly in polynomial-time by using the ellipsoid methods, but a more practical solution is provided by an algorithm which takes a follow-the-perturbed-leader approach. Our experiments illustrate the fact that this approach can successfully learn robust policies from imperfect expert data, in both discrete and continuous environments. It will be interesting to see whether our maxmin formulation can be combined with other methods in RL such as hierarchical learning to produce robust solutions in larger problems.

## 7 Acknowledgement

Doina Precup and Jessie Huang gratefully acknowledge funding from Open Philanthropy Fund and NSERC which made this research possible. Yang Cai and Fa Wu thank the NSERC for its support through the Discovery grant RGPIN2015-06127 and FRQNT for its support through the grant 2017-NC-198956.

## Footnotes

[1]Note that the safety as used here is more in the context of AI safety, and a policy is safe because it is robust to misspecified rewards and the consequent negative side effects.

[2]See Section 3 for the formal definition of consistent rewards. Intuitively, it means that the expert's policy yields a reward that is within $\epsilon$ of the optimal possible reward.

[3]Note that with a polynomial number of trajectories, one can apply standard Chernoff bounds to derive such inequalities that hold with high probability. It is often the case that the probability is so close to 1 that the inequality can be treated as true always for any practical purposes.

[4]Note that $SO_F$ only depends on the bit complexity of $M\backslash R$, but not the actual model of $M\backslash R$ such as the distributions of the initial states or the transition probabilities. We only require access to $ALG$ and an upper bound of $\langle M\backslash R \rangle$.

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
