[Supplementary Material · Learning Safe Policies with Expert Guidance FULL with Supplementary.pdf]

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

# Supplementary Material

## 8 Missing Proofs from Section 4

**The Ellipsoid Method** The following theorem, reworded from [15, 9, 14], states that given a separation oracle of a convex polytope, the ellipsoid method can optimize any linear function over the convex polytope in polynomial time.

**Theorem 3** (Ellipsoid Method). *([15, 9, 14]) Let $P$ be a $d$-dimensional closed, convex subset of $\mathbb{R}^d$ defined as the intersection of finitely many halfspaces, and SO be a poly-time separation oracle for $P$. Then it is possible to find an element in $argmax_{x \in P}\{c \cdot x\}$ for any $c \in \mathbb{R}^d$ (i.e. solve linear programs) in time polynomial in $d$ and $\langle P \rangle$ using the ellipsoid method, if $P$ can be described implicitly using $\langle P \rangle$ bits.* [5]

*Proof of Lemma 2:* Lemma 4 shows that $P_F$ can be implicitly described using $\langle P_F \rangle = \text{poly}(\langle M \setminus R \rangle, \langle \phi \rangle, k)$ bits. Maximizing any linear function $w \cdot \mu$ can be solved by querying ALG on MDP $M \setminus R$ with reward function $w \cdot \phi(\cdot)$. Since MDP $(M \setminus R, w \cdot \phi(\cdot))$ has bit complexity polynomial in $\langle M \setminus R \rangle$, $\langle \phi \rangle$, $k$, and $\langle w \rangle$, we can solve the linear optimization problem in time $\text{poly}(\langle P_F \rangle, k, \langle w \rangle)$. By Lemma 1, we can solve the separation problem in time $\text{poly}(\langle P_F \rangle, k, \langle y \rangle)$ on any input $y \in \mathbb{R}^k$. Hence, we can design a polynomial time separation oracle. $\square$

**Lemma 4.** *Polytope $P_F$ for any MDP without reward function $M \setminus R$ can be implicitly described using $\text{poly}(\langle M \setminus R \rangle, \langle \phi \rangle, k)$ bits.*

*Proof.* The following constraints explicitly describe all $\mu \in P_F$, where $x_{sa}$s correspond to the occupancy measure of some policy $\pi$.

$$\mu = \sum_s \phi(s) \sum_a x_{sa}$$

$$\sum_a x_{sa} = \Pr(s_0 = s) + \gamma \sum_{s',a} x_{s'a} P_{sa} \qquad \forall s$$

$$x_{sa} \geq 0$$

Our statement follows from the fact that all the coefficients in these constraints have bit complexity $\langle M \setminus R \rangle$ or $\langle \phi \rangle$. $\square$

**Intuition behind Lemma 3** The intuition behind Lemma 3 is that the separation oracle $SO_F$ tries to search over all possible weights $w$ to find one to separate the query point $\mu$ from $P_F$ using the ellipsoid method. Along the way, it queries a set of weights (this is our set $C$) on ALG trying to find a separating weight $w$ such that $\mu \cdot w > \mu_w \cdot w$. If such a separating weight is found, $SO_F$ terminates immediately and outputs "No" together with the corresponding separating hyperplane. The $SO_F$ says "Yes" only when it has searched over a polynomial number of weights and concludes that there is no possible weight to separate $\mu$. The reason that $SO_F$ can draw such a conclusion is due to the ellipsoid method. In particular, when $SO_F$ says "Yes", the correctness of the ellipsoid algorithm implies that $\mu$ is in the convex hull of all the extreme points of $P_F$ that have been outputted by the ALG.

**Follow-the-Perturbed-Leader** Kalai and Vempala [13] proposed the FPL algorithm and showed that in expectation, the regret is small against any oblivious adversary. [11] showed that the same regret bound extends to settings with adaptive adversary. To obtain a high probability bound, one can construct a martingale to connect the actual reward and the expected reward obtained by the agent, then apply the Hoeffding-Azuma inequality.

**Theorem 4** (Follow-the-Perturbed-Leader). *[13, 11, 8] Let $d_1, \ldots, d_T$ be a sequence of decisions. Let $s_1, \ldots, s_T$ be a state sequence chosen by an adaptive adversary, that is, $s_t$ can be selected based on all the previous states $s_1, \ldots, s_{t-1}$ and all the previous decisions $d_1, \ldots, d_{t-1}$ for every $t \leq T$.*

*If we let $d_t$ be $\mathrm{argmax}_{d \in D} d \cdot \left( \sum_{i=1}^{t-1} s_i + p_t \right)$, where $p_t$ is drawn uniformly from $[0, 1/\delta]^n$ for some $\delta > 0$, then*

$$\mathbb{E} \left[ \sum_{t=1}^{T} d_t \cdot s_t - \max_{d \in D} \sum_{t=1}^{T} d \cdot s_t \right] \geq -\delta \cdot C_1 C_2 T - \frac{2C_3}{\delta}.$$

*$C_1$ is an upper bound of $||s||_1$ for all $s \in \mathbb{S}$, $C_2$ is an upper bound of $|d \cdot s|$ for all $d \in D$ and $s \in \mathbb{S}$, and $C_3$ is an upper bound of $||d||_1$ for all $d \in D$. Moreover, for all $\xi \geq 0$, with probability at least $1 - \xi$, the actual accumulative reward under any adaptive adversary satisfies,*

$$\sum_{t=1}^{T} d_t \cdot s_t - \max_{d \in D} \sum_{t=1}^{T} d \cdot s_t \geq -\delta \cdot C_1 C_2 T - \frac{2C_3}{\delta} - 2C_2 \sqrt{T \ln \frac{1}{\xi}}.$$

Using Theorem 4, we are ready to prove Theorem 2.

*Proof of Theorem 2:* We use $P$ to denote the sequence $p_1, \ldots, p_T$ and $Q$ to denote the sequence $q_1, \ldots, q_T$. First, notice that every realization of $Q$ defines a deterministic adaptive adversary for the agent. In the setting of Algorithm 3, we can take $C_1$ to be $k$, $C_2$ to be $k^2$, and $C_3$ to be $k$. By Theorem 4 (Section 8 of the supplementary material), we know that for all $\xi \geq 0$, $\mathrm{Pr}_{P \sim U[0,1/\delta]^{kT}}[\sum_{t=1}^{T} \mu_t \cdot w_t - \max_{\mu \in P_F} \sum_{t=1}^{T} \mu \cdot w_t \geq -k^2 \sqrt{T}(3 + 2\sqrt{\ln 1/\xi}) | Q] \geq 1 - \xi$ for every realization of $Q$. Similarly, every realization of $P$ also defines a deterministic adaptive adversary for the designer, and by Theorem 4, we know that $\mathrm{Pr}_{Q \sim U[0,1/\delta]^{kT}}[-\sum_{t=1}^{T} \mu_t \cdot w_t + \min_{w \in P_R} \sum_{t=1}^{T} \mu_t \cdot w \geq -k^2 \sqrt{T}(3 + 2\sqrt{\ln 1/\xi}) | P] \geq 1 - \xi$ for any realization of $P$. Let $B = k^2 \sqrt{T} \left( 3 + 2\sqrt{\ln 1/\xi} \right)$. By the union bound, with probability at least $1 - 2\xi$ over the randomness of $P$ and $Q$

$$\sum_{t=1}^{T} \mu_t \cdot w_t - \max_{\mu \in P_F} \sum_{t=1}^{T} \mu \cdot w_t \geq -B \tag{1}$$

and

$$-\sum_{t=1}^{T} \mu_t \cdot w_t + \min_{w \in P_R} \sum_{t=1}^{T} \mu_t \cdot w \geq -B \tag{2}$$

Next, we argue that $\frac{1}{T} \cdot \sum_{t=1}^{T} \pi_t$ is an approximate maxmin policy.

$$\min_{w \in P_R} \sum_{t=1}^{T} \mu_t \cdot w \geq \sum_{t=1}^{T} \mu_t \cdot w_t - B \qquad \text{(Eq. (2))}$$

$$\geq \max_{\mu \in P_F} \sum_{t=1}^{T} \mu \cdot w_t - 2B \qquad \text{(Eq. (1))}$$

$$\geq T \cdot \max_{\mu \in P_F} \min_{w \in P_R} \mu \cdot w - 2B$$

The last inequality is because that on the LHS (line 2) the designer is choosing a fixed strategy $\frac{1}{T} \cdot \sum_{t=1}^{T} w_t$, while on the RHS (line 3) the designer can choose the worst possible strategy for the agent. Therefore, if the agent uses policy $\frac{1}{T} \cdot \sum_{t=1}^{T} \pi_t$, it guarantees expected reward $\max_{\mu \in P_F} \min_{w \in P_R} \mu \cdot w - 2B/T$.

Finally, in every iteration $t$, we query ALG once to compute $\pi_t$ and $\mu_t$, and we use the ellipsoid method to find $w_t$ using $O(k^2(\langle M \backslash R \rangle^2 + (\log T)^2))$ queries to $SO_R$ and $\mathrm{poly}(k, \langle M \backslash R \rangle, \log T)$ regular computation steps. During each query, $SO_R$ calls ALG. Thus, our result is a reduction from the maxmin learning problem to simply solving an MDP under given weights. Any improvement on ALG will also improve the running time of Algorithm 3. We discuss the empirical running time in section 10 of the supplementary material. □

## 9 Maxmin Learning using an Approximate MDP Solver

In the previous sections, we assume that we have access to an MDP solver ALG that solves any MDP $M$ optimally in time polynomial in $\langle M \rangle$. However, in practice, solving large-size MDPs,

e.g. continuous control problems, exactly could be computationally expensive or infeasible. Our FPL-based algorithm also works in cases where we can only solve MDPs approximately.

Suppose we are given access to an additive FPTAS $\widehat{\text{ALG}}$ for solving MDPS. More specifically, $\widehat{\text{ALG}}$ finds in time polynomial in $\langle M \rangle, 1/\eta$ a solution $(\pi_\eta^*, \mu_\eta^*)$, such that $\mathbb{E}_{s_0 \sim D}[V^{\pi_\eta^*}(s_0)|M] \geq \max_\pi \mathbb{E}_{s_0 \sim D}[V^\pi(s_0)|M] - \eta$. Notice that the weights of $M$'s reward function have $L_1$-norm $L$.

We face two challenges when we replace ALG with $\widehat{\text{ALG}}$: (i) we can no longer find the best policy $\mu_t$ with respect to all the previous weights plus the perturbation in every iteration, and (ii) we no longer have a separation oracle for $P_R$, as the $SO_R$ (Algorithm 1) relies on the MDP solver when $P_R$ is implicitly specified by the expert's policy. It turns out (i) is not hard to deal with, as the FPL algorithm is robust enough to work with only an approximate leader. (ii) is much more subtle. We design a new algorithm and use it as a proxy for the polytope $P_R$. We call this new algorithm a weird separation oracle (following the terminology in [7]) as the points it may accept do not necessarily form a convex set, even though it does accept all points in $P_R$. It may seem at first not clear at all why such a weird separation oracle can help us. However, we manage to prove that just with this weird separation oracle, we can still compute an approximate minimizing weight vector $w_t$ in $P_R$ in every iteration (Step 5 of Algorithm 3). Combining this with our solution for challenge (i), we can still compute an approximately maxmin policy with essentially the same performance as in Algorithm 3.

**Theorem 5.** *If we replace the exact MDP solver* ALG *with an approximate solver* $\widehat{\text{ALG}}$ *in step 4 of Algorithm 3, then for any $\xi \in (0, 1/2)$ and any $c > 0$, with probability at least $1 - 2\xi$, Algorithm 3 finds a policy $\pi$ after $T$ rounds of iterations such that its expected reward under any weight from $P_R$ is at least $\max_{\mu \in P_F} \min_{w \in P_R} \mu \cdot w - \frac{k^2 \left( 6 + 4\sqrt{\ln 1/\xi} \right)}{\sqrt{T}} - 2c$. In every iteration, Algorithm 3 makes one query to $\widehat{\text{ALG}}$ and a polynomial number of queries to $SO_R$. In particular, for every query to $\widehat{\text{ALG}}$, we first divide the input by $2T$ then feed it to $\widehat{\text{ALG}}$ and ask for a policy that is at most $c/2T$ worse than the optimal one.*

The proof of Theorem 5 is similar to the proof of Theorem 2. We use the bounds provided by Lemma 5 instead of Theorem 4, and change the RHS in Equation (1) from $-B$ to $-k^2 \sqrt{T} \left( 3 + 2\sqrt{\ln 1/\xi} \right) - 2cT$ accordingly. The rest of the proof remains the same.

Assume we have a procedure $M_\eta$ for $\eta$-approximating linear programs over the decision set $D$ such that for all $s \in \mathbb{R}^k$,
$$s \cdot M_\eta(s) \geq \text{argmax}_{d \in D} s \cdot d - \eta.$$

**Lemma 5** (Follow the Approximate Perturbed Leader). *[4] Let $d_1, \ldots, d_T$ be a sequence of decision made by an $\eta$-approximating procedure $M_\eta$ such that $d_t = M_\eta(\sum_{i=1}^{t-1} s_i + p_t)$. Then*

$$\mathbb{E}\left[ \sum_{t=1}^T d_t \cdot s_t - \max_{d \in D} \sum_{t=1}^T d \cdot s_t \right] \geq -\delta \cdot C_1 C_2 T - \frac{2C_3}{\delta} - 2\eta T.$$

*The definition of constants $C_1$, $C_2$ and $C_3$ are the same as in Theorem 4. Moreover, for all $\xi \geq 0$, with probability at least $1 - \xi$, the actual accumulative reward under any adaptive adversary satisfies,*

$$\sum_{t=1}^T d_t \cdot s_t - \max_{d \in D} \sum_{t=1}^T d \cdot s_t \geq$$
$$- \delta \cdot C_1 C_2 T - \frac{2C_3}{\delta} - 2C_2 \sqrt{T \ln \frac{1}{\xi}} - 2\eta T.$$

An astute reader may have noticed that in the analysis above, we used the same separation oracle $SO_R$ as in section 3. However, in the case when the separation oracle for the reward polytope is implicitly specified by an expert policy, $SO_R$ queries the MDP solver in step 1 of algorithm 1. If we do not have an exact MDP solver ALG, it is not clear how we can define a separation oracle for polytope $P_R$. We use Algorithm 4 as an proxy to polytope $P_R$.

We call $WSO_R^\eta$ a weird separation oracle for for the reward polytope for $P_R$, because the set of $w'$ that it will accept is not necessarily convex. For example, the following may happen. First, we query

---

**Algorithm 4** Weird Separation Oracle $WSO_R^\eta$ for the reward polytope $P_R$

---

1: Let $\mu_{w'}^{(\eta)} := \widehat{\mathrm{ALG}}(w', \eta)$. It is the feature vector of the policy computed by the approximate MDP solver $\widehat{\mathrm{ALG}}$ with accuracy $\eta$, and $\mu_{w'}^{(\eta)} \cdot w' \geq \max_{\mu \in P_R} \mu \cdot w' - \eta$.

2: **if** $\mu_{w'}^{(\eta)} \cdot w' > \mu_E \cdot w' + \epsilon$ **then**

3:     output "NO" , and $\left(\mu_E - \mu_{w'}^{(\eta)}\right) \cdot w + \epsilon \geq 0$ as the separating hyperplane, since for all $w \in P_R, \mu_E \cdot w \geq \mu_{w'}^{(\eta)} \cdot w - \epsilon$.

4: **else**

5:     output "YES".

6: **end if**

---

two points $w_1$ and $w_2$ that are close to each other. Both are accepted by $WSO_R^\eta$, and it happens to be the case that $\widehat{\mathrm{ALG}}(w_1, \eta)$ and $\mathrm{ALG}(w_2, \eta)$ are both $\eta$ away from the optimal solutions. Now we query $w_3 = (w_1 + w_2)/2$, and run $WSO_R^\eta$. Luckily (or unfortunately) $\widehat{\mathrm{ALG}}(w_3, \eta)$ is close to optimal, and $w_3$ is rejected.

**Lemma 6.** *For any linear optimization problem, we can construct a polynomial time algorithm based on the ellipsoid-method that queries $WSO_R^\eta$, such that it finds a solution that is at least as good as the best solution in polytope $P = \{w | w \cdot \mu_E \geq w \cdot \widehat{\mathrm{ALG}}(w', \eta) - \epsilon, \forall w', \|w'\|_1 \leq L\}$, although our solution does not necessarily lie in $P$.*

*Proof of Lemma 6:* We only sketch the proof here. Solving a linear optimization can be converted into solving a sequence of feasibility problems by doing binary search on the objective value. We show that for any objective value $\alpha$, as long as there is a solution $x \in P$ whose objective value $c \cdot x \geq \alpha$, our algorithm also finds a solution $x'$ such that $c \cdot x' \geq \alpha$. First, imagine we have a separation oracle for $P$, and the ellipsoid method needs to run $N$ iterations to determine whether there is a solution in $P$ whose objective value is at least $\alpha$. The correctness of ellipsoid method guarantees that if it hasn't found any solution after $N$ iterations, then the intersection of the halfspace $c \cdot x \geq \alpha$ and $P$ is empty. The reason is that if the intersection is not empty it must have volume at least $r$, and the ellipsoid method maintains an ellipsoid that contains the intersection of the halfspace $c \cdot x \geq \alpha$ and $P$ and shrinks the volume of the ellipsoid in every iteration. After $N$ iterations the ellipsoid already has volume less than $r$.

Our algorithm also runs the ellipsoid method for $N$ iterations. In each iteration, we first check the constraint $c \cdot x \geq \alpha$, if not satisfied, we output this constraint as the separating hyperplane. If it is satisfied, instead of querying the real separation oracle for $P$, we query $WSO_R^\eta$. If the answer is "YES", we have found a solution $x$ such that $c \cdot x \geq \alpha$. If the answer is "NO", clearly this query point is not in $P$, and the outputted separating hyperplane contains the intersection of the halfspace $c \cdot x \geq \alpha$ and $P$. Therefore, whenever our algorithm accepts a point, it must have objective value higher than $\alpha$. Otherwise, the shrinking ellipsoid still contains the intersection of the halfspace $c \cdot x \geq \alpha$ and $P$. If our algorithm terminates after $N$ iterations without accepting point, we know that the intersection between the halfspace $c \cdot x \geq \alpha$ and $P$ is empty as the volume of the ellipsoid after $N$ iterations is already too small.

$\square$

Consider the following three polytopes:

(i) $P_R := \{w \mid w \cdot \mu_E \geq w \cdot \mu - \epsilon, \ \forall \mu \in P_F\}$

(ii) $P = \{w | w \cdot \mu_E \geq w \cdot \widehat{\mathrm{ALG}}(w', \eta) - \epsilon, \forall w', \|w'\|_1 \leq L\}$

(iii) $P_R^{(\epsilon+\eta)} := \{w \mid w \cdot \mu_E \geq w \cdot \mu - \epsilon - \eta, \forall \mu \in P_F\}$.

**Fact 1.** $P_R \subseteq P$.

**Fact 2.** $WSO_R^\eta$ only accepts points that are in $P_R^{(\epsilon+\eta)}$.

*Proof.* Suppose $w \notin P_R^{(\epsilon+\eta)}$, then clearly $w \cdot \widehat{\mathrm{ALG}}(w, \eta) \geq \max_{\mu \in P_F} w \cdot \mu - \epsilon - \eta > w \cdot \mu_E$. Hence, $WSO_R^\eta$ will not accept $w$. $\qquad\square$

**Lemma 7.** *For all $w$ in $P_R^{(\epsilon+\eta)}$, $w \cdot \frac{\epsilon}{\epsilon+\eta}$ is in $P_R$.*

*Proof of Lemma 7:* From the definition of $P_R^{(\epsilon+\eta)}$, multiply both side of the inequality with $\frac{\epsilon}{\epsilon+\eta}$, and let $w' = w \cdot \frac{\epsilon}{\epsilon+\eta}$, $w'$ is in $P_R$. $\square$

**Theorem 6.** *For any $c > 0$ and $\xi \in (0, 1/2)$, with probability at least $1 - 2\xi$, Algorithm 3 finds a policy $\pi$ after $T$ rounds of iterations such that its expected reward under any weight from $P_R$ is at least $\max_{\mu \in P_F} \min_{w \in P_R} \mu \cdot w - \frac{k^2 \left(6 + 4\sqrt{\ln 1/\xi}\right)}{\sqrt{T}} - 4c$. In every iteration, Algorithm 5 makes one query to $\mathrm{ALG}_\eta$ and a polynomial number of queries to Algorithm 4.*

Now, we are ready to describe the algorithm using only access to $WSO_R^\eta$.

---

**Algorithm 5** Finding the Maxmin Policy using Follow-the-Perturbed-Leader (FPL)

---

**input** $T$: the number of iterations
1: Set $\delta := \frac{1}{k\sqrt{T}}$, where $||w||_1 \leq L$ for all $w \in P_R$. Set $\eta_1 := \frac{c}{2T}$ and $\eta_2 := \frac{c\epsilon}{2k^2T - c}$.
2: Arbitrarily pick some policy $\pi_1$ and compute $\mu_1 \in P_F$. Arbitrarily pick some reward weights $w_1$, and set $t = 1$.
3: **while** $t \leq T$ **do**
4:     Let policy $\pi_t$ and $\mu_t = \Psi(\pi_t)$ be the output of $\widehat{\mathrm{ALG}}\left(\left(\sum_{i=1}^{t-1} w_i + p_t\right)/T, \eta_1\right)$, where $p_t$ is drawn uniformly from $[0, 1/\delta]^k$.
5:     Use our algorithm in Lemma 6 with $WSO_R^{\eta_2}$ to solve $\min w^T (\sum_{i=1}^{t-1} \mu_t + q_t)$, where $q_t$ is drawn uniformly from $[0, 1/\delta]^k$. Let $w'_t$ be the solution and set $w_t$ to be $w'_t \cdot \frac{\epsilon}{\epsilon+\eta_2}$.
6:     $t := t + 1$.
7: **end while**
8: Output the randomized policy $\frac{1}{T} \cdot \sum_{t=1}^T \pi_t$.

---

*Proof of Theorem 6:* At each time step $t$, using $WSO_R$, Algorithm 5 step 5 outputs a $w_t$. By Lemma 6 and Fact 1,

$$w'_t \cdot \left(\sum_{i=1}^{t-1} \mu_i + q_t\right) \leq \min_{w \in P_R} w \cdot \left(\sum_{i=1}^{t-1} \mu_i + q_t\right).$$

By Lemma 7 and Fact 2,

$$w_t \cdot \left(\sum_{i=1}^{t-1} \mu_i + q_t\right) = \frac{\epsilon}{\epsilon + \eta_2} \cdot w'_t \cdot \left(\sum_{i=1}^{t-1} \mu_i + q_t\right)$$

$$\leq \min_{w \in P_R} w \cdot \left(\sum_{i=1}^{t-1} \mu_i + q_t\right) + \frac{2k^2 T}{\epsilon + \eta_2},$$

where we used the fact that

$$-w_t \left(\sum_{i=1}^{t-1} \mu_i + q_t\right) \leq 2k^2 T.$$

Since $c = \frac{2\eta_2 k^2 T}{\epsilon + \eta_2}$, we can use Lemma 5 and replace the RHS in Equation (2) that was used in the proof of Theorem 2 to $-k^2 \sqrt{T} \left(3 + 2\sqrt{\ln 1/\xi}\right) - 2cT$. The analysis for $\mu_t$ remains the same as in Thoerem 5. $\square$

# 10 Experiment Details

In every iteration of Algorithm 3 and Algorithm 5, step 5 computes a minimizing weight in $P_R$. Instead of using the ellipsoid method to solve the LP, we use the analytic center cutting-plane method (see [5] for a brief overview) throughout our experiments. The method combines good practical performance with reasonable simplicity.

## 10.1 Gridworld

The domain contains five types of terrain. Four terrain types are used in the demonstration gridworld where we construct the expert policy. We select the rewards for these four terrain types uniformly from $[-0.5, 0]$, and the target has a reward of 10. The reward of each terrain type is deterministic. The demonstration MDP is uniformly composed of four terrain types, 25% each type. The fifth terrain type (red colored as in Figure 3) is not present in the demonstration gridworld. The agent is trained in a "real-world" MDP that is composed uniformly of all five terrain types, 20% each. We select maps that are feasible, such that for all rewards in the consistent reward polytope, value iteration has a solution for the agent to reach the goal. We use feature vectors that indicate the terrain type of each state, choose a discount factor of 0.95, and use value iteration throughout the experiment. The consistency between the expert policy and the reward function is defined with $\epsilon = 0.5$.

**Deterministic transition model** In an MDP with deterministic transition model, the agent moves in exactly the direction chosen by the agent. We run FPL for $5000$ iterations and use the average of policies output by the last 2500 iterations as the maxmin policy. Figure 2 shows that our maxmin policy is much safer than a baseline. The baseline policy is computed in an MDP whose reward weights are the same as the demonstration MDP for the first four terrain types and the fifth terrain weight is chosen uniformly at random from [-1,0]. The expert policy for the displayed results is constructed by computing the optimal policy in an demonstration MDP with rewards for the first four terrain type set as $[-0.5, -0.2, -0.4, -0.1]$. The results are accumulated from 100 individual runs using the same expert policy. Examples of the baseline trajectories are shown in Figure 7.

Figure 7: Trajectories chosen by policies generated using weights randomly assigned to the red-colored unknown feature. Although this feature may have negative side effects, the random agent may still go through it.

**Stochastic transition model** At each state, there is 10% chance that the agent will go in a random direction regardless of the action chosen by the agent. The agent will receive rewards based on the state it actually lands in. We show in Figure 8 that to mitigate the higher risk of traversing the unknown terrain type, our maxmin policy appears to be more conservative than the deterministic case. Although Figure 9 shows that it cannot absolutely avoid the unknown terrain type due of the stochastic nature of the model, the percentage is much lower than the baseline. The baseline was computed with the same reward weights as in the deterministic case.

**Computation Performance** In our grid world experiment, the worst case running time of ALG is $O(n^2)$, but experiments show a more benign runtime of $O(n^{1.5})$. For a $50 \times 50$ grid world with

Figure 8: At each state, there is $10\%$ chance that the agent will go in a random direction irrespective of the action chosen by the agent, our maxmin method is still valid. Comparing to Figure 3 (**right**), the maxmin policy also avoids going to the peripheral of the red-colored unknown feature.

Figure 9: In the gridworld with stochastic transition model, our maxmin policy has a small chance of traversing the unknown terrain type disregard of being more conservative than the maxmin policy in the deterministic case. The percentage is much lower than the baseline.

25 features, Algorithm 3 appears to converge after 325 iterations of FPL with total runtime of 3324 seconds (average of 20 trials, ordinary desktop computer). Instead of using the ellipsoid method, we used analytic center cutting-plane method, and the running time appears to scale in the order of $O(k^2)$.

## 10.2  CartPole

We modify the classic CartPole task in the OpenAI Gym environment by adding features that may incur additional rewards. This is represented by the question blocks in Figure 4. The two question blocks correspond to feature indicators for the agent's horizontal position in the range of $[-1.2, 0)$ and $[0.6, 1.8)$. We keep the same episode termination criterions for the pole angle and cart position as the original environment. An episode is considered ending without failing if the pole angel and cart position meet the criterion and the episode length is greater than 500. The agent receives a reward of $+1$ for surviving every step.

We use longer episodes than the original problem to allow more diverse movement, while it also makes the task more challenging. During validation of a policy, we consider the task solved as getting a target average reward over 100 consecutive episodes with less than five failed episodes. The target average reward depends on the reward we assign for passing the question blocks. If each step spent at question block $i$ incurs reward of $r_i$, the target average reward is set to be $450 + 25 \sum r_i$. For example, in $scenario\ A$, only the blue question block exists and it incurs reward of $-2$, our expert policy $Expert\ A$ passes the validation criterion by getting average reward higher than 400 over 100 consecutive episodes with less than 5 failed episodes. Indeed, our $Expert\ A$ policy performs quite well by getting a reward greater than 450 in $scenario\ A$. In $scenario\ B$, only the yellow question block is present and it incurs reward of $+2$. $Expert\ B$ passes the validation criterion with reward greater than 1000.

The agent is in an MDP with both blue and yellow question blocks whose reward polytopes are implicitly defined by the expert policy. We use Q-learning and apply updates using minibatches of stored samples as the MDP solver. Notice that for this problem, our MDP solver is not necessarily optimal. We computed maxmin policies when provided with different expert policies. The results in Figure 6 are from testing the maxmin policy for 2000 episodes.