[Reviews · NeurIPS 2018]

Reviewer 1



================ Light review ================ I am not an expert in this area, but I think the results presented in this work are accurate. Some questions and comments for the authors below: 1. The paper is very hard to follow. It would be good to describe the model/problem first and then the solution. As it stands, especially in the introduction, the problem statement and solution concept are mixed up. 2. The use of "safe policies" in this work is quite misleading. The technical/research connotation of safety in the context of RL/control/robotics is regarding exploration -- does the agent avoid visiting parts of the state space that are unsafe/unstable. Safety as used here is in the context of AI safety, which I feel is largely a non-technical or popular press use of the term. I would encourage the authors to phrase the problem differently (e.g. robustness to misaligned goals), to avoid confusion in the research community. 3. While the connections to robust optimization literature is interesting, it is unclear what the broader impact of the work would be. The assumptions used are very restrictive, largely to be able to directly borrow results from optimization literature. The algorithms/approaches proposed in this work also crucially rely on these restrictive assumptions (not just for proofs) and hence cannot be applied more broadly. Thus, I would encourage the authors to have a section on how to extend these ideas to cases where the assumptions don't hold -- can any insights be carried over? It would also help to have experiments on more difficult/elaborate tasks.

Reviewer 2



Learning from demonstrations usually faces an ill-posed problem of inferring the expert reward functions. To facilitate safe learning from demonstrations, the paper formulates a maximin learning problem over a convex reward polytope, in order to guarantee that the worst possible consistent reward would yield a policy that is not much worse than optimal. The assumption is that the reward is linear in known features. The authors proposed two method: (i) ellipsoid method and (ii) follow-the-perturbed leader using separation oracles and a given MDP solver. The experiment is done in a grid world setting, and a modified version of the cart-pole problem. One piece that I failed to understand is from line 61-63: why apprenticeship learning won't return the same policy as the proposed solution? This seems like a major point for me, as the formulation looks very similar to Syed & Schapire 2008 (although they used Multiplicative Weights Update as the online learning procedure. There they also used linear reward assumption, and reduction of policy learning to feature expectation matching, similar to Abbeel & Ng 2004). Since the contrast to previous method was not discussed beyond this paragraph, I'm curious what the comparison would look like in the experiment? It seems that this discussion is missing from the experiment section. I think the paper will benefit from at least some additional intuition in the experiment / discussion sections into what's wrong with Syed & Schapire method (or also Apprenticeship Learning via Linear Programming, where iteratively invoking the oracles is not needed). The other concern, as perhaps is also clear to the authors and other readers, is the scalability of the algorithms. Repeatedly invoking separation oracle and MDP solver may mean that this method will have limited applicability to higher dimensionalities. Gridworld experiment: How many environments did you use for training? How many demonstrations? Cartpole experiment: how well would the method work without the addition of the two question blocks? This is a relatively minor point, but I think the focus of the paper has a lot more to do with robust policy learning (due to worst case guarantee) than safe learning, which to me implies safe exploration / satisfying safety criteria during training in addition to high reward learning. The paper is nicely written and well-organized.

Reviewer 3



Summary: The paper proposes a framework for learning safe policies in the setting where reward functions are unknown but a set of expert demonstrations are available. Under the assumption of linear reward functions, the set of expert demonstrations are used to form a polytope of all reward functions that are consistent with expert. Then to ensure safety, the paper propose to perform maxmin optimization: learning a policy to maximize the worst possible reward from the reward polytope. Under the availability of separation oracles and MDP solvers, the paper proposed two algorithms: 1 an exact ellipsoid-based method and (2) an algorithm the uses FTPL to alternatively updating policy and reward function to approximately solve the maxmin problem. The authors also showed how the maxmin formulation they used is different from previous imitation learning framework. strengths: the paper is well-wrriten and the algorithms and theorems are clear. Comments: 1. The FPL maxmin learning requires ellipsoid method with access to SO_R to solve the argmin problem to find the reward function (maybe it should be clearly stated in Alg 3 or the text around Alg 3). As the query on SO_F and query on SO_R both use ALG as internal solver, is it possible to explicitly work out the complexity of both algorithm in terms of the number of ALG calls to show the gap between the ellipsoid method and Alg 3?

Reviewer 4



This paper considers a type of safe RL where 1) the rewards are unknown + not seen 2) there are expert demonstrations The setting assumes nature can set the weights on reward features such that they are consistent with the expert demos, but that the selected policy will do as poorly as possible. The goal is therefore to maximize the minimum return of the policy. This also results in the agent being able to learn in environments different from the demo environments: any states that correspond to new reward features, since they could be arbitrarily bad, are avoided. The paper's contributions are primarily theoretical, but I do like that it shows at least some experimental results in two toy domains. There are some details missing from the experimental section making the results impossible to replicate. For example, how many experts were used in the experiments? How many demonstrations were given? How close to optimal were they? Section 2 defines an MDP as having a finite number of states, but experiments include a continuous state task. This should be clarified. Similarly, when talking about an algorithm that "can find the optimal policy for an MDP in polynomial time" it would help to say exactly what it is polynomial in. Overall the paper is well written. The sentence on line 120 appears to be missing a verb.